# Functionalization of Morin-Loaded PLGA Nanoparticles with Phenylalanine Dipeptide Targeting the Brain

**DOI:** 10.3390/pharmaceutics14112348

**Published:** 2022-10-31

**Authors:** Mario Alonso, Emilia Barcia, Juan-Francisco González, Consuelo Montejo, Luis García-García, Mónica-Carolina Villa-Hermosilla, Sofía Negro, Ana-Isabel Fraguas-Sánchez, Ana Fernández-Carballido

**Affiliations:** 1Department of Pharmaceutics and Food Technology, School of Pharmacy, Universidad Complutense de Madrid, Plaza de Ramón y Cajal s/n, 28040 Madrid, Spain; 2Institute of Industrial Pharmacy, School of Pharmacy, Universidad Complutense de Madrid, Plaza de Ramón y Cajal s/n, 28040 Madrid, Spain; 3Department of Chemistry in Pharmaceutical Sciences, School of Pharmacy, Universidad Complutense de Madrid, Plaza de Ramón y Cajal s/n, 28040 Madrid, Spain; 4Department of Health and Pharmaceutical Sciences, School of Pharmacy, Universidad San Pablo-CEU, 28668 Boadilla del Monte, Spain; 5Department of Pharmacology, Pharmacognosy and Botany, School of Pharmacy, Universidad Complutense de Madrid, Plaza de Ramón y Cajal s/n, 28040 Madrid, Spain; 6Brain Mapping Lab, Pluridisciplinary Research Institute, Universidad Complutense de Madrid, Ciudad Universitaria s/n, 28040 Madrid, Spain

**Keywords:** Alzheimer’s disease, morin hydrate, PLGA, nanoparticles, blood–brain barrier, rhodamine B

## Abstract

Alzheimer’s disease (AD) is the most prevalent neurodegenerative disorder, with its incidence constantly increasing. To date, there is no cure for the disease, with a need for new and effective treatments. Morin hydrate (MH) is a naturally occurring flavonoid of the *Moraceae* family with antioxidant and anti-inflammatory properties; however, the blood–brain barrier (BBB) prevents this flavonoid from reaching the CNS when aiming to potentially treat AD. Seeking to use the LAT-1 transporter present in the BBB, a nanoparticle (NPs) formulation loaded with MH and functionalized with phenylalanine-phenylalanine dipeptide was developed (NPphe-MH) and compared to non-functionalized NPs (NP-MH). In addition, two formulations were prepared using rhodamine B (Rh-B) as a fluorescent dye (NPphe-Rh and NP-Rh) to study their biodistribution and ability to cross the BBB. Functionalization of PLGA NPs resulted in high encapsulation efficiencies for both MH and Rh-B. Studies conducted in Wistar rats showed that the presence of phenylalanine dipeptide in the NPs modified their biodistribution profiles, making them more attractive for both liver and lungs, whereas non-functionalized NPs were predominantly distributed to the spleen. Formulation NPphe-Rh remained in the brain for at least 2 h after administration.

## 1. Introduction

Alzheimer’s disease (AD) is the most prevalent neurodegenerative disorder, with its incidence constantly increasing as elderly populations grow worldwide. Currently, more than 55 million people live with this disease worldwide, with nearly 10 million new cases every year [1]. It is characterized by progressive memory loss and cognitive dysfunction. Its etiology remains unclear, although it is likely to be related to both genetic and environmental factors. To date, there is no cure for the disease, and the goals of treatment are to slow down the progress of neurodegeneration and maintain, as far as possible, the quality of life of AD patients. New therapies should concentrate on increasing neuronal survival by avoiding the main mechanisms of cell death occurring in AD: hyperphosphorylation of the tau protein and formation of amyloid-β peptide aggregations (Aβ) [2,3].

Morin hydrate (MH) is a naturally occurring flavonoid of the *Moraceae* family with antioxidant and anti-inflammatory properties that could be a potential candidate for the treatment of neurodegenerative diseases such as Parkinson’s disease (PD) [4] and Alzheimer’s disease (AD) [5]. Its potential use in AD could be based on several mechanisms of action, as MH has proven to decrease the formation of Aβ plaques in mice models [6,7], also reducing hyperphosphorylated tau protein levels in neurons at concentrations around 1–2 µM [8,9]. Both mechanisms are involved in neuronal cell death occurring in AD [10]. In addition, MH could act as a neuroprotective agent by inhibiting NF-κβ pathways and preventing neuronal apoptosis [11]. However, the practical use of MH is currently limited mainly due to its poor water solubility and very low oral bioavailability (<1%). MH has a very short elimination half-life of around 30 min and suffers extensive first-pass hepatic metabolism [12,13]. In addition, MH exhibits limited permeability through physiological barriers such as the blood–brain barrier (BBB) [14,15].

Nanosystems such as polymeric nanoparticles (NPs) prepared with PLGA are one of the approaches used to facilitate the passage of drugs across the BBB [16,17,18], and to avoid early metabolization [19]. To increase the access of drugs to the CNS, NPs can be functionalized with surfactants or ligands. In this regard, different molecules, including peptides, have been linked to the NPs to improve their affinity for the brain [20,21,22].

The BBB is a complex and efficient barrier that prevents foreign substances from entering the CNS. Small endogenous molecules such as amino acids, glucose, and others that are necessary for the survival of brain cells, use specific transporters expressed at the luminal side of the endothelial cells present in the BBB [23]. One of these protein transporters is LAT-1 (Large Aminoacid Transporter-1) which is primarily expressed in brain, thymus, spleen, skeletal muscle, and several cancer cells [24]. Various studies have shown that gabapentin and L-Dopa, which are amino acid-like drugs, are transported across the BBB via LAT-1 [25,26]. Gonzalez-Carter et al. [27] demonstrated that L-DOPA-functionalized gold nanoparticles efficiently and specifically crossed the BBB via the LAT-1 protein. Moreover, the choroid plexus is the location of the blood–cerebrospinal fluid barrier, with the peptide transporter PepT2 expressed at this barrier. This endogenous transporter is mainly located in the kidneys, brain, and lungs, being responsible for peptide transport through this barrier [28].

Phenylalanine (phe) is a naturally occurring amino acid that can be found in proteins. As a substratum of LAT-1, phenylalanine may be an interesting option as a targeting ligand for the development of polymeric NPs destined to facilitate the passage of drugs across the BBB when treating CNS disorders [29,30].

In this work we aimed to develop and characterize a new formulation of biodegradable MH nanoparticles functionalized with phenylalanine (phe-phe) dipeptide to facilitate their passage across the BBB and remain in the CNS for as long as possible. For this, two types of MH-loaded NPs were developed using PLGA as polymer: non-functionalized MH-loaded PLGA NPs and PLGA:PLGA-phe-phe-functionalized MH-loaded NPs. To study their biodistribution and ability to cross the BBB in rats, rhodamine B (Rh-B), a fluorescent dye unable to cross the BBB, was used for the development of Rh-B-loaded PLGA NPs and PLGA:PLGA-phe-phe-functionalized Rh-B-loaded NPs.

## 2. Materials and Methods

### 2.1. Materials

Morin hydrate (MH), rhodamine B (Rh-B), phenylalanine dipeptide (phe-phe), N-hidroxysuccinimide (NHS) and 1-ethyl-3-(3-dimethylaminopropyl)-carbodiimide (EDC) were obtained from Sigma-Aldrich Química, S.A. (Madrid, Spain). PLGA Resomer^®^ RG 502 with a ratio of 50:50 poly (D.L-lactic-co-glycolic) acid (Mw 12,000 Da) was obtained from Evonik Industries AG (Essen, Germany). Dichloromethane (DCM), polyvinyl alcohol (PVA, Mw 30,000–70,000 Da) and dimethyl sulfoxide (DMSO) were provided by Sigma-Aldrich Química, S.A. (Madrid, Spain). Distilled and deionized water (Q-POD^®^ Milli-Q system, Millipore, Madrid, Spain) was used in the preparation of all buffers and solutions. For ^1^H NMR analysis all reagents were purchased from Sigma-Aldrich Química, S.A. (Madrid, Spain) and Fluka (Madrid, Spain). Solvents were obtained from SDS (Madrid, Spain). All solvents and reagents were of analytical grade and were used as received without further purification.

### 2.2. Synthesis of PLGA-phe-phe

#### 2.2.1. Activation of PLGA

PLGA-COOH (2.5 g, 0.21 mmol) was dissolved in dry DCM (7 mL), converted to PLGA-NHS with an excess of N-hydroxysuccinimide (NHS, 68 mg, 0.6 mmol) in the presence of 1-ethyl-3-(3-dimethylaminopropyl)-carbodiimide (EDC, 115 mg, 0.6 mmol), and then left to stirred for 22 h at room temperature under argon atmosphere. The resulting solution was added dropwise to ice-cooled ethyl ether (10 mL), and the solid was decanted and repeatedly washed in an ice-cooled mixture of ethyl ether and methanol (10 mL, 1/1, *v*/*v*) to remove residual by-products. The solid was diluted with DCM (15 mL), dried under anhydrous sodium sulfate, and filtered. The solvent was removed under vacuum in a room-temperature water bath. Finally, the solid was freeze-dried (Lyo Quest^®^, Telsta Technologies S.L., Madrid, Spain) to obtain PLGA-NHS (2.2 g) as a white powder. The spectrum obtained was ^1^H NMR (250 MHz, Chloroform-d) δ 5.33–5.16 (m, 11H), 4.97–4.71 (m, 26H), 2.86 (d, J = 5.0 Hz, 4H), and 1.62 (d, J = 7.0 Hz, 61H).

#### 2.2.2. Conjugation of Phe-Phe Peptide to Activated PLGA

PLGA-NHS (502 mg, 0.041 mmol) and dipeptide phe-phe (Mw 311 g/mol, 18.6 mg, 0.06 mmol) were dissolved in 12 mL of dry DCM and then N, N-diisopropylethylamine (0.01 mL, 0.064 mmol) was added. The mixture was gently stirred for 40 h under an argon atmosphere. The crude reaction was concentrated to dryness under vacuum without external heating. The resulting solution was washed with 15 mL of ice-cooled petroleum ether/methanol co-solvent (10/90, *v*/*v*). The solid was decanted and repeatedly washed with the same solvent to remove unreacted peptides and residual by-products. Finally, the solid was freeze-dried to obtain PLGA-phe-phe (120 mg) as a white powder. The spectrum obtained was ^1^H NMR (250 MHz, Chloroform-d) δ 7.17 (m, 10H), 5.20 (d, J = 7.2 Hz, 110H), 4.82 (s, 200H), 1.58 (d, J = 6.8 Hz, 950H), and 1.28 (d, J = 6.6 Hz, 750H).

Reactions were monitored by thin layer chromatography in 0.20 mm silica gel 60 F254 plates (Merck, Madrid, Spain) with fluorescent indicator (SDS CCM221254) and visualized with a UV lamp. For proton magnetic resonance spectroscopy (H1 NMR) spectra were acquired using a Bruker Avance 250 spectrometer working at 250 MHz for 1H and 63 MHz for 13C and operated via the standard Bruker software (Version 4.1, Nuclear Magnetic Resonance, Centre for Research Assistance, UCM, Madrid, Spain). Chemical shifts are reported as parts per million (ppm) relative to tetramethylsilane, and spin multiplicities are given as s (singlet), d (doublet), t (triplet), q (quartet), or m (multiplet).

### 2.3. Elaboration of Nanoparticles

#### 2.3.1. Elaboration of Morin Hydrate/Rhodamine B-loaded PLGA Nanoparticles

Morin-loaded PLGA nanoparticles (Table 1, formulation NP-MH) were prepared by the solvent extraction-evaporation method using a DCM:water system. Briefly, 50 mg of PLGA 502 was dissolved in 2 mL of DCM. An exact amount of MH (5 mg) was dissolved in DMSO (100 µL) and added to the DCM solution. The resulting mixture was added dropwise into 6 mL of 1.5% PVA solution. Both phases were sonicated at 80% amplitude for 10 min in pulses (15 s on, 10 s off). The organic solvent was removed under magnetic stirring for 4 h. The NPs were washed with water and centrifuged at 13,000 rpm for 30 min (Sorvall ST 8R centrifuge, Thermo Scientific, Waltham, MA, USA). Finally, the NPs were resuspended in 1 mL of 1.5% sucrose solution and freeze-dried for 24 h.

Rhodamine B-loaded PLGA NPs (formulation NP-Rh) were also prepared by the same method and used in biodistribution studies. Briefly, 2.5 mg of Rh-B and 50 mg of PLGA 502 were dissolved in DCM (2 mL). This solution was added dropwise into 6 mL of 1.5% PVA and sonicated. The organic solvent was removed. The NPs were then washed, centrifuged, and freeze-dried under the same conditions. 

Blank PLGA NPs were also prepared (formulation NP-0). All formulations were prepared in triplicate.

#### 2.3.2. Elaboration of Morin Hydrate/Rhodamine B-Loaded PLGA-Phe-Phe Nanoparticles

Formulation NPphe-MH was prepared as NP-MH using 50 mg of PLGA:PLGA-phe-phe (60:40 ratio) instead of PLGA (Table 1), which was dissolved in DCM (2 mL). MH (5 mg) was dissolved in DMSO (100 µL) to obtain the organic phase. This solution was added dropwise into 6 mL of 1.5% PVA and sonicated for 10 min in pulses (15 s on, 10 s off). The emulsion formed was stirred for 4 h to remove the organic solvent. Finally, the NPs were washed, centrifuged, and freeze-dried under the same conditions. 

Formulation NPphe-Rh was prepared as NP-Rh using PLGA:PLGA-phe-phe (60:40 ratio) instead of PLGA (Table 1). For this, 30 mg of PLGA 502, 20 mg of PLGA-phe-phe, and 2.5 mg of Rh-B were dissolved in DCM (2 mL). This solution was added dropwise into the PVA solution and then sonicated. Finally, DCM was removed, and the NPs recovered and washed before being freeze-dried. In addition, NPs were prepared without MH/Rh-B (NPphe-0). All formulations were prepared in triplicate.

### 2.4. Characterization of Nanoparticles

#### 2.4.1. Morphology

The shape and surface morphology of the NPs were analyzed by scanning electron microscopy (SEM, Jeol JSM 7600F, Jeol Ltd., Tokyo, Japan). Samples were coated with a thin layer of colloidal gold applied in a cathodic vacuum evaporator before observation in a microscope.

The mean diameter and size distribution of the NPs were determined by laser diffraction using a Zetatrac^®^ Ultra 3500 system (Microtrac MRB, Montgomeryville, PA, USA). The diameter was expressed as volume diameter. The polydispersity index (PDI) was also calculated.

#### 2.4.2. Zeta Potential

Zeta potential was determined by Laser-Doppler anemometry using a Malvern Zetasizer Nano S^®^ (Malvern Panalytical, Malvern, U.K.). Measurements were carried out at 25 °C in aqueous solution. For this, 5 mg of each formulation was suspended in 50 mL of distilled water and stirred for 1 min. Then, the aqueous dispersion of NPs at a concentration of 100 µg/mL was placed in a capillary cell (Cell Enhances Capillary^®^, Malvern Panalytical, Malvern, U.K.) for zeta potential measurements. All formulations were analyzed in triplicate.

#### 2.4.3. Encapsulation Efficiency

Encapsulation efficiency (EE%) of MH within the NPs was determined as follows: 10 mg of each formulation was dissolved in 1 mL of DCM. Then, 15 mL of methanol was added to this solution and centrifuged for 5 min at 5000 rpm. The supernatant (5 mL) was mixed with 5 mL of HCl 0.1 N and analyzed by HPLC [31]. The equipment used consisted of a Jasco chromatograph (Jasco International Co., Ltd., Tokyo, Japan) equipped with a PU-2080 pump and a UV/Vis 2070 detector. The chromatographic column selected was Gemini 5 μm NX-C18 (110 Å, 250 × 4.6 mm) (Phenomenex, Madrid, Spain). The mobile phase consisted of acetonitrile:water (27:73, *v*/*v*). The flow rate was 1.4 mL/min, and the volume injected was 50 µL. The detection wavelength was 256 nm. All analyses were carried out at 40 ± 0.5 °C. Drug loading (DL) was also determined by HPLC and expressed as the amount of MH incorporated by 100 mg of NPs.

To determine the EE% and DL of Rh-B, the same procedure was used. Quantification was performed by spectrophotometry at 555 nm.

#### 2.4.4. In Vitro Release Studies

In vitro release studies were carried out in a Memmert WB22 water bath (Memmert, Schwabach, Germany) at 37 ± 1 °C and constant agitation (100 rpm). For this, 20 mg of NPs were suspended in PBS at pH 7.4 (5 mL). At regular time intervals, samples were centrifuged with the supernatant withdrawn and filtered through 0.45 µm filters. Quantification of MH was performed by HPLC [31]. Quantification of Rh-B was conducted by spectrophotometry at 555 nm. All in vitro release tests were performed in triplicate.

### 2.5. Biodistribution Studies

Male Wistar rats (Harlan France SARL, Gannat, France) weighing 220–270 g were housed with controlled temperature and a 12 h:12 h light:dark cycle, observed daily, and fed with a commercial pelleted diet and water ad libitum. All animal procedures were approved by the Ethics Committee on Animal Experimentation (Universidad Complutense de Madrid, UCM) and the regional authorities (Madrid Region, PROEX 14/18). All procedures complied with the European Community Council Directive (010/63/UE). Efforts were made to minimize the number of animals used and their suffering.

The animals were divided into three groups: -Group 1 (*n* = 4) received saline solution.-Group 2 (*n* = 8) received formulation NP-Rh.-Group 3 (*n* = 6) received formulation NPphe-Rh.

The NPs were suspended in saline at a concentration of 25 mg/mL. The volume administered was normalized by weight using 0.5 mL as a reference volume for the average animal weight. The suspension was injected into the rat tail vein using a 30-gauge needle. Finally, the rats were anesthetized with isoflurane. Half of the animals within each group were sacrificed at time 1 h and the other half at time 2 h.

#### 2.5.1. Organ Biodistribution

After sacrifice, the liver, lungs, kidneys, and spleen were extracted from all the animals. Brains were also extracted and divided into left and right hemispheres (one hemisphere was used for the quantification of Rh-B and the other one for the biodistribution study of NPs). All organs were kept at −80 °C prior to analysis. 

To observe the biodistribution of NPs prepared with Rh-B, a portion (1.5 ± 0.1 g) of each organ (liver, kidney, spleen, lung) was homogenized using a turbine with 5 mL of DCM. After 1 h, the mixture was centrifuged at 8000 rpm for 10 min to extract all Rh-B content. Rh-B was quantified by measuring its fluorescence in a Varian Cary Eclipse fluorescence spectrophotometer (Agilent Technologies, Madrid, Spain). The procedure used was that described by Marcianes et al. [32] and adapted to our experimental conditions. Excitation wavelength set at 555 nm and emission at 570 nm. The fluorescence intensity from non-treated animals was used as a background signal (negative controls) to discard tissue autofluorescence. 

The amount of Rh-B was expressed as ng of Rh-B/g of tissue. As the different formulations showed different drug loading, the results were normalized for the same dose of Rh-B.

#### 2.5.2. Passage of NPs through the BBB

To study the passage of NPs through the BBB, brains were divided into both hemispheres. The right hemispheres were sectioned into 25 µm slices, embedded for 24 h in DPX (dibutylphenylphtalate (10 mL) + polystyrene (25 g) + xylene (70 mL)), stained with DAPI solution (4′,6-diaminidino-2-pheylindole) (125 ng/mL) [33], and analyzed by confocal microscopy in a Leica microscope (Leica Microsystems GmbH, Wetzlar, Germany) at 545 nm excitation and 570 nm emission [34]. Quantification of fluorescence intensity was performed with the ImageJ software [35]. Fluorescence intensity from control brains was used as a background signal (negative controls) to discard tissue autofluorescence. 

The left hemispheres were homogenized with DCM (5 mL) and centrifuged at 8000 rpm for 10 min to extract all Rh-B content. The amount of Rh-B was expressed as ng of Rh-B/g of tissue.

#### 2.5.3. Statistical Analysis

Results were expressed as mean ± standard deviation. Multifactorial ANOVA tests were performed using the Statgraphics 19-X64^®^ Centurion software (Statgraphics Technologies, Inc., The Plains, VA, USA). A *p*-value < 0.05 was considered statistically significant.

## 3. Results and Discussion

### 3.1. Synthesis of PLGA phe-phe

The surface modification of nanosystems can be carried out for several purposes: to improve the stability of polymeric NPs, increase their residence time within the body, reduce aggregation, improve their passage through biological barriers, and/or achieve cellular internalization [36]. 

At the level of the BBB, multiple transporters are expressed, influencing the barrier permeability for various small molecules, such as amino acids, glucose, and iron transferrin [37]. In this regard, amino acids, short peptides, polypeptides, proteins, and their combinations are gaining special attention for the design of complex structures on the nanometric scale due to their structural diversity, biocompatibility, and scalability, among other properties [38,39].

On the other hand, amyloid fibrils are the hallmark of several diseases, including Alzheimer’s disease [40,41]. Reches and Gazit [42] conducted a study in which small molecules containing aromatic residues (diphenylalanine peptide) were self-assembled into well-ordered nanotubes being able to control the formation of Aβ aggregates. 

For these reasons, we have developed a novel nanosystem in the form of polymeric NPs functionalized with phenylalanine dipeptide (phe-phe). When attaching surface ligands to nanosystems, some authors have indicated that functionalization of the polymer prior to the elaboration of the NPs is quite effective as it allows for adequate control of the surface decoration [43].

In our study, the -CO2H groups of PLGA were activated with 1-ethyl-3-(3-dimethylaminopropyl) carbodiimide (EDC) and N-hydroxysuccinimide (NHS). The activated polymer was purified, with activation being confirmed by proton magnetic resonance spectroscopy (H1 NMR). Then, activated PLGA reacted with the phe-phe peptide, and functionalization was confirmed by H1 NMR with the presence of aromatic signals at 7.2 ppm and missing characteristic NHS group signals at 2.9 ppm. The conjugation efficiency obtained was around 80% (calculated as the percentage conversion of the reactant with respect to the product) (Figure 1).

### 3.2. Characterization of Nanoparticles

As shown in Figure 2, all NPs formulations were spherical in shape with smooth surfaces. Laser diffraction revealed that NPs had sizes around 190 nm (Table 2). PLGA-phe-phe NPs resulted in slightly larger particle sizes than PLGA NPs, although statistically significant differences (*p* < 0.05) were only found when compared to unloaded NPs. Particle sizes under 200 nm will allow the nanosystems to cross the BBB [44,45,46]. Therefore, our NPs formulations could be appropriate for CNS drug delivery.

All formulations presented PDI ≤ 0.3 (Table 2), indicating low polydispersity in particle size [47].

All formulations exhibited negative zeta potential values. Particles with zeta potentials of more than +30 mV or more than −30 mV are normally considered stable with a low tendency for aggregation. Positively charged NPs may cause adverse systemic effects, such as hemolysis. On the contrary, negative or neutral charges appear to show longer circulating times and fewer side effects [48].

When comparing zeta potential values of blank NPs (NP-0 vs. NPphe-0), the effect of the presence of phenylalanine was clear. Formulation NP-0 exhibited a value lower than −20 mV (−25.4 ± 1.4 mV), as expected due to the carboxylic acid groups present in PLGA. Other authors have also obtained similar results [49,50]. Regarding NPphe-0, a zeta potential value of −18.3 ± 2.2 mV was obtained, probably due to the presence of the dipeptide and the reduction of free carboxylic acid groups in PLGA (Table 2).

Differences in zeta potential were also found between non-functionalized and phe-phe functionalized MH-loaded NPs. In NP-MH, the flavonoid increased the zeta potential to −20.8 ± 0.5 mV, which was slightly lower in NPphe-MH (−23.3 ± 0.7 mV). This could be explained by the fact that in NPphe-MH, not all the carboxylic acid groups were expressed on the surface of the particles. If we compare the values of EE%, functionalization of NP-MH with the dipeptide (formulation NPphe-MH) led to higher encapsulating efficiency, as some of the phe moieties could be interacting with the flavonoid, favoring its affinity for the polymeric matrix, thereby leading to higher EE%. If so, these phe moieties would not be present on the surface of the NPs.

From these findings, it would be expected to achieve similar results with the Rh-B formulations. However, Rh-B-loaded NPs (NP-Rh and NPphe-Rh) showed the greatest differences in zeta potential (−22.6 ± 0.4 mV vs. −7.88 ± 0.2 mV). For NPphe-Rh, zeta potential markedly increased. Rh-B is positively charged, and the EE% of NPphe-Rh was two-fold that of NP-Rh. The presence of the fluorescent dye on the surface of the NPs could account for this variability. This is supported by data obtained from the in vitro release tests in which formulation NPphe-Rh exhibited the highest release of the fluorescent dye within the first hour.

High EE% values (>80%) were obtained for all MH-loaded NPs and especially for formulation NPphe-MH with a mean value of 96.5 ± 3.2% (Table 2). Similar EE% was obtained by other authors [51] when encapsulating MH within PLGA NPs elaborated by a solid-in-oil-in-water (S/O/W) emulsification method, and in which a suspension of MH was prepared in the organic phase prior to its addition to the PVA solution.

Regarding EE% of Rh-B within PLGA NPs, low values were obtained (<20%); however, formulation NPphe-Rh resulted in two-fold EE% than NP-Rh. This may be explained by the fact that Rh-B is freely dissolved in water. 

The higher EE% found in both MH- and Rh B-loaded functionalized PLGA NPs may be attributed to the interaction of aromatic phe moieties present in the polymer with aromatic rings of MH and Rh-B, which favors their affinity for the polymeric matrix, thereby increasing EE% of both agents. 

In vitro release tests led to similar profiles during the first 24 h for non-functionalized and functionalized MH-loaded NPs (Figure 3). Mean burst releases (1 h) were 45.51 ± 1.84% and 47.17 ± 3.15% for formulations NP-MH and NPphe-MH, respectively. 

Thereafter, the zero-order release kinetics of MH was observed for both formulations up to 192 h. During this time, NPs prepared with the amino acid (NPphe-MH) showed a slightly faster release of the flavonoid. With this formulation, therapeutic concentrations could be theoretically maintained for 200 h. In fact, the therapeutic concentrations of MH when potentially treating AD do not necessarily need to be high. Alberdi et al. [8] found that cell death caused by Aβ oligomers in neuronal cultures was reduced in the presence of micromolar concentrations of MH (1 µmol), resulting in attenuated oxidative stress. 

In vitro release tests were also conducted for Rh-B-loaded NPs to clarify the release behavior of Rh-B during the first two hours, taking into consideration that these formulations have been only developed for the in vivo biodistribution and BBB studies. As these studies have been carried out 1 h and 2 h after administration, our aim was to evaluate the in vitro release of Rh-B from the NPs at these times. Mean Rh-B release percentages of 44.00 ± 0.35% and 55.04 ± 0.44% were obtained for formulations NP-Rh and NPphe-Rh, respectively, at time 2 h. 

Biodistribution studies were performed for both Rh-B-loaded PLGA NPs (NP-Rh and NPphe-Rh), with fluorescence being detected in the liver, kidneys, spleen, and lungs of Wistar rats. Multifactorial ANOVA tests were applied to the data considering experimental variables formulation, organ biodistribution, and time. From the results obtained, statistically significant differences in organ distribution were found (*p*-value = 0.005). In addition, statistically significant differences resulted between both formulations (*p*-value = 0.005) in all the tissues analyzed. As non-statistically significant differences were found when both times (1 h and 2 h) were analyzed together with the other variables, additional independent ANOVA tests were carried out to analyze the effects of time, organ distribution, and formulation (Figure 4).

The results obtained at 1 h showed that formulation NP-Rh preferentially accumulated in the spleen. Formulation NPphe-Rh exhibited higher affinity for the liver and lungs than for kidneys and spleen. 

At time 2 h, the accumulation of formulation NP-Rh in the spleen was significantly reduced with respect to 1 h (*p*-value < 0.05). Regarding NPphe-Rh, the greatest differences in biodistribution were found in the liver and lungs when comparing both times. Indeed, a statistically significant (*p*-value < 0.05) increase in the liver and decrease in the lungs was found at 2 h when compared to 1 h. As occurred at time 1 h, the non-functionalized NPs preferentially accumulated in the spleen, followed by the liver and lungs, whereas functionalized NPs showed higher affinity for the liver and lungs and low affinity for the spleen.

It is known that PLGA NPs mainly accumulate in the liver and spleen after i.v administration due to sequestration by cells of the mononuclear phagocytic system (MPS), with particle size and surface characteristics being key parameters influencing their biodistribution [36]. Mohammad and Reineke [52] also found that suberoylanilide hydroxamic acid (SAHA)-loaded PLGA NPs extensively distributed throughout the organs of the MPS with NPs preferentially found in the liver and kidneys, then followed by the heart and lungs.

In our study, the liver was the organ of highest deposition, followed by the spleen and lungs (Figure 2 and Figure 3). This result agrees with the fact that NPs with hydrophobic surfaces are rapidly sequestered from the circulation by organs of the MPS. 

In our study, the effect of endogenous transporters on the biodistribution of NPs was not analyzed. As particles sizes of NP-Rh and NPphe-Rh were similar, differences found in biodistribution could be, at least partially, attributed to differences in their surfaces, as the presence of phenylalanine in the polymer matrix might modify the biodistribution of NPs at the studied time points.

To investigate the ability of NPs to cross the BBB, brains were removed after sacrifice and divided into both hemispheres. Images obtained from one of the hemispheres clearly showed the presence of Rh-B (Figure 5), even though free Rh-B cannot cross the BBB [53].

In addition, the quantification of the fluorescence intensity of Rh-B in brain samples was performed using ImageJ software (Figure 6A). At time 1 h both formulations (NP-Rh and NPphe-Rh) presented similar values (*p*-value > 0.05). However, at 2 h, the intensity of fluorescence from formulation NPphe-Rh was significantly higher. This might indicate that the presence of the dipeptide (phe-phe) in the formulation increased the passage of NPs through the BBB at longer times or, at least, prevented them from being expelled by means of the P-glycoprotein (P-gp) efflux pump, a transport mechanism, which is usually involved [32].

Figure 6B shows the mean amounts (±SD) of Rh-B determined in brain samples (left hemispheres) by means of fluorescence spectrophotometry. Results are expressed as ng of Rh-B/g of tissue, being like those obtained by ImageJ software. At time 1 h, both formulations resulted in similar amounts of Rh-B without statistically significant differences found between them (*p*-value > 0.05). However, at time 2 h, the amount of Rh-B was significantly higher for formulation NPphe-Rh in comparison to NP-Rh. 

The similar results obtained by both analytical techniques (microscopy and fluorometry) seem to indicate that attaching phenylalanine to the NPs surfaces increased their residence time within the CNS. It is known that this amino acid shows a favorable gradient to access the brain without being expelled [54], which could explain the behavior observed with the functionalized formulation (NPphe-Rh) in this organ.

The results obtained at 2 h are very encouraging, taking into consideration that when developing nanosystems targeting the CNS, the activity of the P-gp efflux pump is a drawback that needs to be overcome. Rh-B is a known P-gp substrate; therefore, once released from the NPs, it will be expelled from the brain. For this, the results obtained indicate that either the NPphe-Rh NPs crossed more slowly the BBB but to a greater extent than NP-Rh or remained in the brain for a longer period while releasing Rh-B, whereas non-functionalized NPs would have been already expelled by the efflux pumps.

Limitations of our research include the small number of animals tested, high interindividual variability, and the difficulty of predicting the in vivo release of MH from the results obtained in the in vitro release tests conducted with the NPs. 

## 4. Conclusions

The nanoparticulate formulations developed were adequate for encapsulating MH flavonoid; however, when using the phenylalanine dipeptide in the elaboration of NPs, an increase in the encapsulation efficiency of MH and Rh-B was obtained. As both molecules possess aromatic rings in their structures, the increases obtained may be attributed to the interaction between these aromatic rings and that of phenylalanine, leading to a higher affinity of MH and Rh-B for the polymeric matrix of the NPs. 

The presence of phenylalanine dipeptide might have contributed to modifying the biodistribution profile of the NPs, making them more attractive for the liver and lungs. On the contrary, non-functionalized NPs were predominantly distributed to the spleen. Nevertheless, more research should be conducted to investigate the effects of endogenous transporters on the biodistribution profiles of the NPs. 

Linkage of phenylalanine dipeptide to PLGA in formulation NPphe-Rh resulted in an increased presence of the NPs in brain tissue at longer times (2 h) which agrees with the affinity of the CNS for the amino acid.

## Figures and Tables

**Figure 1 pharmaceutics-14-02348-f001:**
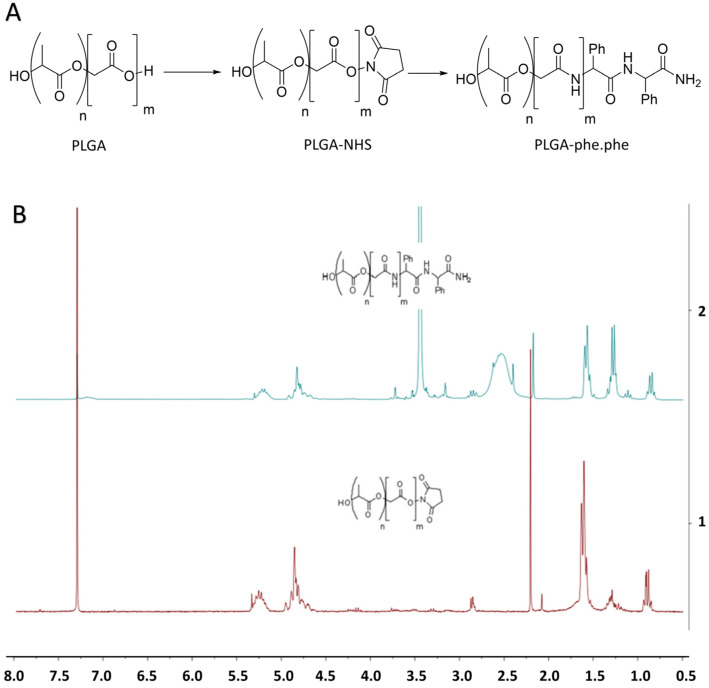
(**A**) Reaction scheme for the activation of carboxylic acid groups in PLGA and its functionalization. (**B**) H1 NMR patterns of PLGA-NHS (upper) and PLGA-phe-phe (lower). NHS: N-hidroxysuccinimide; phe-phe: phenylalanine dipeptide.

**Figure 2 pharmaceutics-14-02348-f002:**
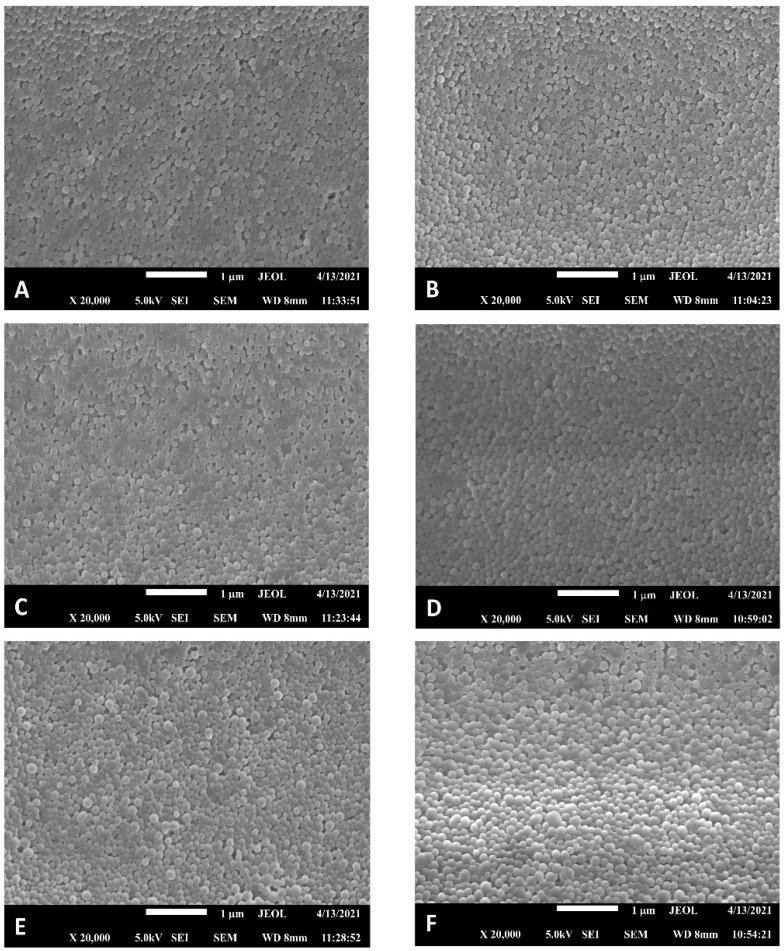
SEM images (x20,000) of all NPs formulations. (**A**) NP-0. (**B**) NPphe-0. (**C**) NP-MH. (**D**) NPphe-MH. (**E**) NP-Rh. (**F**) NPphe-Rh. MH: morin hydrate; phe: phenylalanine; Rh: rhodamine B.

**Figure 3 pharmaceutics-14-02348-f003:**
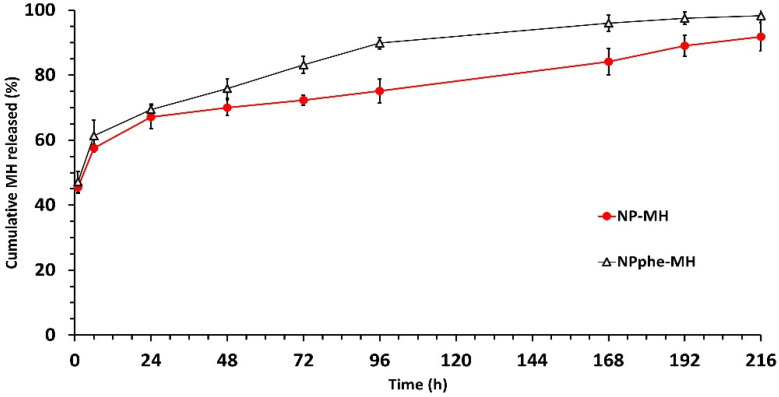
Mean in vitro release profiles (±SD) of MH from formulations NP-MH and NPphe-MH. MH: Morin hydrate; phe: Phenylalanine.

**Figure 4 pharmaceutics-14-02348-f004:**
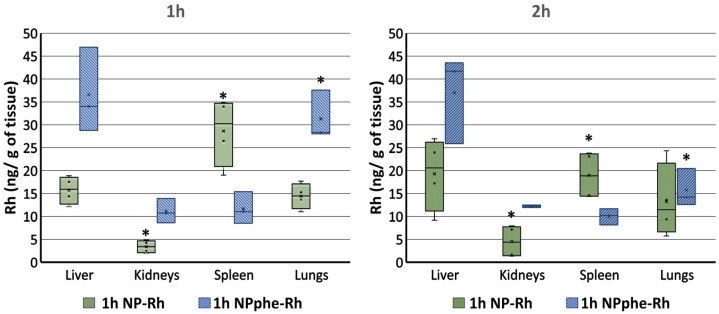
Mean amounts (±SD) of Rh-B in different organs at times 1 h and 2 h. * Statistically significant differences (*p*-value < 0.05).

**Figure 5 pharmaceutics-14-02348-f005:**
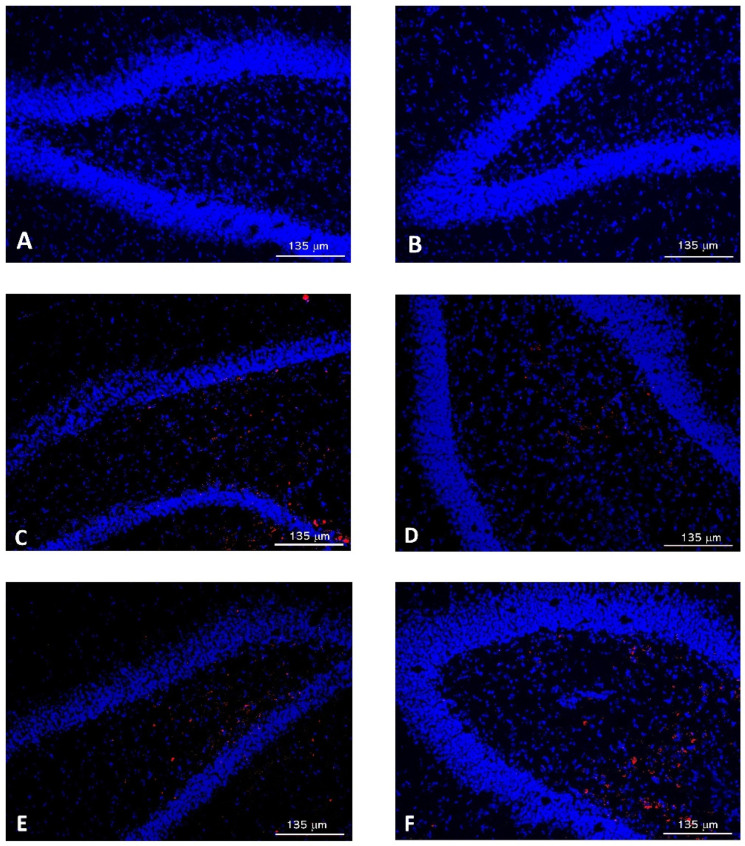
Images of brain samples corresponding to: (**A**) non-treated (control) animals at 1 h; (**B**) non-treated (control) control animals at 2 h; (**C**) formulation NP-Rh at 1 h; (**D**) formulation NP-Rh at 2 h; (**E**) formulation NPphe-Rh at 1 h; (**F**) formulation NPphe-Rh at 2 h.

**Figure 6 pharmaceutics-14-02348-f006:**
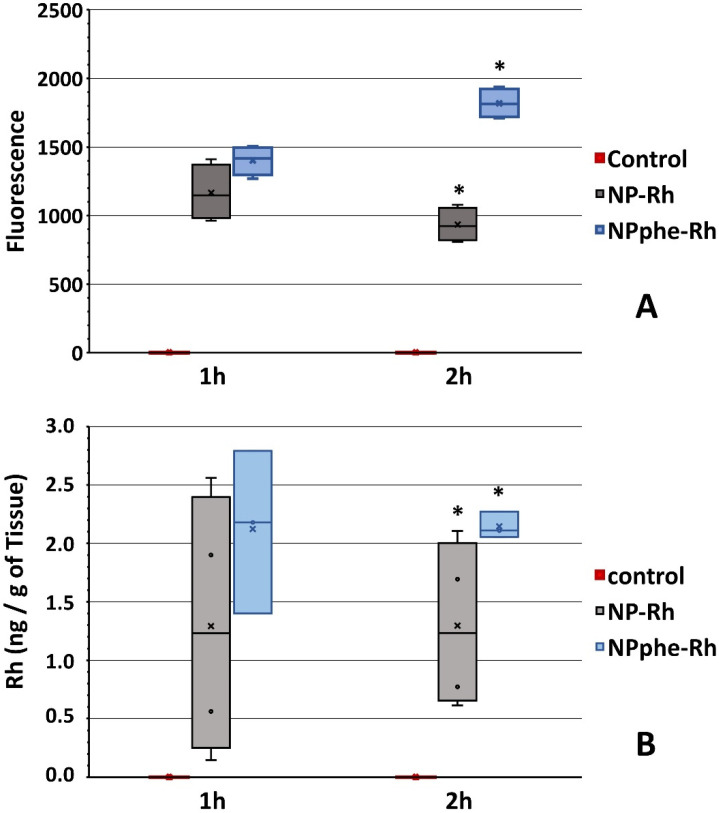
(**A**) Mean fluorescence intensity (±SD) of Rh-B in brain samples at times 1 h and 2 h measured by ImageJ software. (**B**) Mean amounts (±SD) of Rh-B in brain samples at times 1 h and 2 h determined by fluorescence spectrophotometry. Control corresponds to non-treated animals. * Statistically significant differences (*p*-value < 0.05).

**Table 1 pharmaceutics-14-02348-t001:** Nanoparticle (NPs) formulations developed. phe: phenylalanine; MH: morin hydrate; Rh-B: rhodamine B.

Formulation	PLGA (mg)	PLGA-Phe-Phe (mg)	MH (mg)	Rh-B (mg)
NP-0	50	-	-	-
NP-MH	50	-	5	-
NP-Rh	50	-	-	2.5
NPphe-0	30	20	-	-
NPphe-MH	30	20	5	-
NPphe-Rh	30	20	-	2.5

**Table 2 pharmaceutics-14-02348-t002:** Mean (±SD) particle size, zeta potential, drug loading (DL), and encapsulation efficiency (EE%) of the NPs formulations. Mean values of the polydispersity index (PDI).

Formulation	Particle Size ± SD (nm)	PDI	Zeta Potential ± SD (mV)	DL (µg/100 mg NPs) ± SD	EE% ± SD
NP-0	176.4 ± 13.4	0.2	−25.4 ± 1.4	-	
NP-MH	188.6 ± 16.7	0.3	−20.8 ± 0.5	7503.6 ± 88.1	82.5 ± 1.4
NPphe-0	192.8 ± 1.4	0.2	−18.3 ± 2.1	-	
NPphe-MH	189.1 ± 8.2	0.3	−23.3 ± 0.7	9023.8 ± 203.4	96.5 ± 3.2
NP-Rh	187.7 ± 6.9	0.2	−22.6 ± 0.4	362.8 ± 17.3	7.6 ± 0.4
NPphe-Rh	191.0 ± 6.5	0.25	−7.88 ± 0.2	715.3 ± 8.6	15 ± 0.2

## Data Availability

Not applicable.

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
