# Peer review of "Functionalization of Morin-Loaded PLGA Nanoparticles with Phenylalanine Dipeptide Targeting the Brain"

_pharmaceutics, 2022, doi:10.3390/pharmaceutics14112348_

Round 1
Reviewer 1 Report
The manuscript written by Alonso et al. is an interesting study describing the functionalization of PLGA NPs with a dipeptide (composed of two phenylalanine residues) and its impact on the loading efficiency of morin hydrate into NPs, biodistribution profile of NP, and its ability to cross BBB. However, there are some issues that need to be answered/corrected.
The manuscript requires language editing. Also, there are many punctuation mistakes throughout the manuscript, which should be corrected.
Remove dot (.) from the title of the manuscript.
There are many paragraphs in Introduction. Some paragraphs should be merged. This is important to properly narrate the storyline of the manuscript.
There is no quantitative information about the number of dipeptide molecules conjugated to the surface of PLGA nanoparticles. How many dipeptide molecules are conjugated to the surface of NPs?
In section 3.2 of Results, the conjugation efficiency is mentioned to be 80%, meaning that one fifth of the nanoparticles are not conjugated. Is there any way to get rid of these non-conjugated NPs? And how these non-conjugated NPs can negatively impact the data and results reported (characterization of NPs, in vivo biodistribution studies and the ability of NPs to cross BBB)?
Add a reference for this sentence: “…, even though free Rh-B cannot cross the BBB” (line 413).
What are controls in Figures 5 and 6? Describe properly in the legend of the figures.
How do you explain the signification accumulation of NP-Rh and NPphe-Rh in lungs? Only referring to the impact of phenylalanine on changing the surface characteristics of NPs is not enough. What is the mechanism behind this accumulation? Is there any specific receptor reported in the literature that can describe this observation? If there is any specific receptor on the surface of lung cells, why lung accumulation of NPphe-Rh is reduced after 2 hours?
Author Response
See file uploaded

Reviewer 2 Report
The manuscript by Alonso M et al., entitled “Functionalization of Morin-Loaded PLGA Nanoparticles with 2 Phenylalanine Dipeptide Targeting the Brain”. The present work aims to develop and characterize a new formulation of biodegradable MH nanoparticles functionalized with a dipeptide of phenylalanine to facilitate their passage across the BBB and remain in the CNS for as long as possible.
- The introduction could include a paragraph on the epidemiology of the disease with data and reference.
- Must include references to some paragraphs where facts are stated without citing references.
- The average diameter of the nanoparticles was measured, what needs to be shown is the stability of the nanoparticles temporally, as well as how they behave with other media than water. It would be important to be able to add the figure with this data.
- The data in Figure 4 could be converted into a boxplot so it is possible to observe the data and data distribution (better than a histogram). Same as figure 6.
- It needs to be mentioned to put the limitations of the study.
Author Response
Please see file uploaded.

Reviewer 3 Report
The manuscript is well designed and discusses the results well, however some points need to be clarified:
- Please justify the different trial number for each group. How did the authors calculate the sample number?
-The experimental number is very low, considering that the animals will be sacrificed at two different times. At most, the analyzes were performed in quadruplicate. This needs to be justified.
- Why did the authors prefer to subdivide the brains into two areas (hemispheres)? Dosing per brain area provides a more accurate result.
- Why did the authors measure the organs in only two points? Why not increase the experimental n and assess the biodistribution at more points?
Author Response
See file uploaded.

Round 2
Reviewer 2 Report
All requested changes have been made.
Reviewer 3 Report
Questions about the experimental protocol were answered. Now I can be in favor of accepting the manuscript.